# Nutraceutical Supplements in the Thyroid Setting: Health Benefits beyond Basic Nutrition

**DOI:** 10.3390/nu11092214

**Published:** 2019-09-13

**Authors:** Salvatore Benvenga, Ulla Feldt-Rasmussen, Daniela Bonofiglio, Ernest Asamoah

**Affiliations:** 1Department of Clinical and Experimental Medicine-Endocrinology, University of Messina, via Consolare Valeria-Gazzi, 98125 Messina, Italy; s.benvenga@live.it; 2Master Program on Childhood, Adolescent and Women’s Endocrine Health, University of Messina, via Consolare Valeria-Gazzi, 98125 Messina, Italy; 3Interdepartmental Program on Molecular and Clinical Endocrinology and Women’s Endocrine Health, AOU Policlinico G. Martino, via Consolare Valeria-Gazzi, 98125 Messina, Italy; 4Medical Endocrinology and Metabolism PE 2132, Rigshospitalet, Copenhagen University Hospital, Blegdamsvej 9, DK-2100 Copenhagen, Denmark; ufeldt@rh.dk; 5Department of Pharmacy, Health and Nutritional Sciences, University of Calabria, 87036 Arcavacata di Rende (CS), Italy; daniela.bonofiglio@unical.it; 6Community Physicians Network, Diabetes & Endocrinology Care, 8435 Clearvista Place, Suite 101, Indianapolis, IN 46256, USA

**Keywords:** nutraceuticals, thyroid function, dietary supplements

## Abstract

In recent years, there has been a growing interest in nutraceuticals, which may be considered as an efficient, preventive, and therapeutic tool in facing different pathological conditions, including thyroid diseases. Although iodine remains the major nutrient required for the functioning of the thyroid gland, other dietary components play important roles in clinical thyroidology—these include selenium, l-carnitine, myo-inositol, melatonin, and resveratrol—some of which have antioxidant properties. The main concern regarding the appropriate and effective use of nutraceuticals in prevention and treatment is due to the lack of clinical data supporting their efficacy. Another limitation is the discrepancy between the concentration claimed by the label and the real concentration. This paper provides a detailed critical review on the health benefits, beyond basic nutrition, of some popular nutraceutical supplements, with a special focus on their effects on thyroid pathophysiology and aims to distinguish between the truths and myths surrounding the clinical use of such nutraceuticals.

## 1. Introduction

### 1.1. Definition of Nutraceutical

The definition of nutraceuticals is still in the grey area between food, food supplements, and pharmaceuticals. Some definitions [1,2,3,4,5] of nutraceuticals are provided in Table 1. The term “nutraceutical” was coined in 1989 by Stephen De Felice, founder and chairman of the Foundation for Innovation in Medicine, an American organization which encourages medical health research. He defined a nutraceutical as a “food, or parts of a food, that provide medical or health benefits, including the prevention and treatment of disease” [4]. Japan was among the first countries to face the issue of regulating food supplements and foodstuffs. This legislation, originally set in 1991, evolved into the 2003 Health Promotion Law [5]. The current European regulation (Regulation No. 1924/2006 of the European Parliament and of the Council, recently updated by EU Regulation 2015/2283) defines food categories and includes a definition of food supplements, although there is no official mention or recognition the term “nutraceutical” [6]. Accordingly, the European Food Safety Authority (EFSA) does not make any distinction between “food supplements” and “nutraceuticals” for beneficial health claim applications for new products. In a similar way, the Dietary Supplement Health and Education Act (DSHEA, 1994) [7] defined dietary supplements as a category of food, as did the US Food and Drug Administration (FDA) [8]. Indeed, in America “medical foods” and “dietary supplements” are regulatory terms, however “nutraceuticals”, “functional foods”, and other such terms are determined by consultants and marketers, based on consumer trends. Further information on the dietary supplements given by the Food and Drug Administration (FDA) on its website [9] is summarized in Appendix Table A1.

### 1.2. Search of the Literature

A PubMed search, run on 14 July 2017, using the word “nutraceutical” as the entry, yielded 67,344 results. Results fell to 4820 using the entry “nutraceuticals AND hormones” and to 553 using the entry “nutraceuticals AND thyroid”. Approximately 18 months later (5 February 2019), the corresponding numbers were 78,919 (+17%), 5538 (+15%) and 642 (+16%), indicating that the interest in the thyroid proceeds with the same pace as that for nutraceuticals in general and hormones in general. Confirmation of these data came from a final search that was run on 9 July 2019 (Table 2). 

In the following text, different nutraceuticals possibly influencing human thyroid function and/or immunity will be reviewed and commented upon.

A general effect of the nutraceuticals beyond the thyroid effect is not within the scope of this review, nor is a meticulous review of animal or other experimental studies. We were guided by our clinical practices, particularly those for which patients were most curious. As mentioned in the following section, there is indeed a growing market for such nutraceuticals.

There was relatively scant literature on the topic, and most research focused on thyroid cancer and was experimental in nature, concerning the nutraceuticals illustratively mentioned by the Food and Drug Administration, as shown in Appendix Table A1.

### 1.3. Market and Sales

Based on data from a decade ago, annual supplement sales were $23 billion, and about 40,000 supplement products were on the market in the United States [10]. In 2015, the American market for dietary supplements was valued at $37 billion, with the economic impact in the United States for 2016 estimated at $122 billion, including employment wages and taxes [11]. One 2016 analysis estimated the total market for dietary supplements could reach $278 billion worldwide by 2024 [11]. Table 3 summarizes the details for the nutraceuticals reviewed here [12,13,14,15,16].

### 1.4. The Issue of Purity

“The biggest problem with supplements is that many of them do not actually contain what the label claims. As many as 70% of the supplements on the market either don’t have ingredients that match their labels or contain contaminants of some kind” [17]. In his review, Lockwood aimed to investigate the extent of substandard formulated and raw material nutraceuticals [17]. The key findings were that “published evaluations of over 70 formulations of 25 different nutraceuticals revealed variable quality; no nutraceutical showed consistent high quality, but a number revealed consistent low quality, thereby making the case for closer regulation of manufacturers. Whole food sources have also been shown to be widely variable in constituent levels.” [17]. Concerning the issue of purity, the illegal presence of thyroid hormones in the majority of dietary health supplements marketed for “thyroid support” potentially exposes patients to the risk of developing iatrogenic thyrotoxicosis [18].

In the following text, we now give some data concerning the nutraceuticals dealt upon in our paper. Concerning carnitine, of 12 over-the-counter carnitine formulations, the actual mean content was only 52% of that indicated on the label [19]. Furthermore, of the same 12 preparations, five had unsatisfactory pharmaceutical dissolution characteristics.

Concerning myo-inositol, one study evaluated label accuracy of four myo-inositol products, designed for polycystic ovary syndrome (PCOS) treatment and available on the Italian market, and performed a cost comparison based on myo-inositol content in milligrams for products analyzed [20]. A significant difference in the myo-inositol content, compared with the labeling was found for the products. Only one product contained more than 95% of the myo-inositol content claimed on the label, and there was a product with less than 75% of the labeling amount. Based on a 2-g myo-inositol per day dose, the cost of a 30-day supply ranged from Euro 20.77 and Euro 71.86, after correction by the actual amount of myo-inositol.

One recent study aimed to determine the dose of melatonin in food supplements marketed in Europe (pharmacies of Spain) and the United States (supermarkets of San Francisco, CA, USA) by validating a liquid chromatography method with diode array detection (LC-DAD) [21]. The authors tentatively identified eight tryptophan-related contaminants in melatonin supplements, with only one supplement declaring its addition on the label. Label melatonin doses varied from 1–1.95 mg/unit and 0.3–5 mg/unit for supplements marketed in Europe (Spain) and the US, respectively. Four out of 17 supplements showed significant deviations from melatonin content declared on the label (from −60% to −20%). Only five out of the eight supplements purchased in Spain actually met the qualifications needed to claim to reduce the time to fall asleep. Another study analyzed the actual melatonin content (and presence of contaminants) in 31 melatonin supplements purchased from groceries and pharmacies in one city in Canada [22]. Melatonin content varied from −83% to +478% of labeled melatonin and approximately three-fourths had melatonin concentration ≤10% of what was claimed. Worse yet, the content of melatonin between lots of the same product varied by as much as 465%. An additional 26% of the 31 melatonin supplements were found to contain serotonin.

Concerning resveratrol, 14 brands of resveratrol-containing nutraceuticals were evaluated [23]. The 14 preparations were purchased directly from online stores during 2010 and were analysed before their expiry dates. Only five out of 14 brands had near label values, compliant with Good Manufacturing Practices (GMP) requirements (95%–105% content of active constituent), four products were slightly out of this range (83%–111%) and three were in the 8%–64% range. Two samples were below the limit of detection. The greater the difference between actual and labeled resveratrol content, the lower the antioxidant and antiproliferative activity strength.

With regard to selenium, one study analysed six different brands of yeast-based selenium food supplements that were obtained from local stores [24]. These supplements were treated with milder extraction and hydrolysis conditions to analyse for the expected selenomethionine content. Only two brands had high levels of selenomethionine, one brand appeared to contain all inorganic selenium, and one brand appeared to contain greater than half inorganic selenium despite label claims of content being only selenomethionine.

## 2. Carnitine: Compound and Physiology

Carnitine is a quaternary ammonium compound (3-Hydroxy-4-(trimethylazaniumyl) butanoate) that is ubiquitous in tissues and biological fluids of mammals [25]. The natural enantiomer is l-carnitine, which acts as an obligatory cofactor for β-oxidation of fatty acids by facilitating the transport of the long-chain fatty acids across the mitochondrial inner membrane as acyl-carnitine esters. This oxidation liberates energy via the production of ATP in the respiratory chain, thus playing a role in cell’s energy metabolism. Particularly, l-carnitine exerted a physiological benefit with a positive impact on cardiac function through reduced oxidative stress, inflammation and necrosis of cardiac myocytes. [26]. Only 25% of the body stores of carnitine come from biosynthesis and 75% comes from the diet. The main source is red meat and dairy products. Muscles are the most prominent carnitine depository since they store about 95% of the 120 mmol total amount contained in the adult human body, and the concentration in skeletal muscle (3.5 mmol/L) is 70-fold greater than that in plasma.

The main interest in carnitine supplementation comes from athletes and other physical exercise performers [27]. Thus, repeated-dose carnitine supplements may increase skeletal muscle content. For instance, long-distance runners given a daily dose of 2 g carnitine for 28 days and subjected to a four-week training period [28] increased skeletal muscle carnitine by approximately 13% as compared to a decrease of about 10% in placebo-treated athletes. In other athletes, supplementation with 1 g/day carnitine for 120 days of training increased carnitine concentrations in skeletal muscle by an average of 9% compared to a decrease of 5% in the placebo-treated athletes [29]. Carnitine is critical for normal skeletal muscle bioenergetics [30,31,32], and skeletal muscles suffer seriously in states of carnitine deficiency. A relative carnitine deficiency can occur in athletes as a result of increased energy metabolism, unbalanced nutrition, decreased skeletal muscle content and increased renal excretion of carnitine. The important energetic role of carnitine, the relative deficiency associated with sustained physical exercise, and the fact that carnitine is a natural compound, has led healthy subjects aiming to improve their exercise performance to conclude that “more carnitine should be better [30,31,32], but basically this was proven to be without any beneficial effect.

### Carnitine and Thyroid Function

A German group of authors conducted pivotal clinical studies as early as 1959 in a very limited number of patients with Graves’ disease, using a mixture of the two isomers (l-and d-carnitine) [33]. The first patient was a 53-year-old bedridden woman with very severe Graves’ disease and nervousness, insomnia, weight loss, sweating, tachycardia and Graves’ orbitopathy. Basal metabolic rate (BMR) was +82%, and she was administered 1 g/d d,l-carnitine. After 10 days, BMR was unchanged but one week later it fell to +59%. Five weeks after starting d,l-carnitine, BMR was still +50% and the authors switched to the naturally occurring l-carnitine. After only 10 days BMR dropped more rapidly to +8% with associate improvement in general well-being and heart rate. Atrial fibrillation disappeared and heart rate was 80–90 beats/min. To prove that the improvement was due to l-carnitine, it was withdrawn in the 7th week from admission. BMR rose to +39%, but after rechallenge with l-carnitine it fell again to +18% [33].

In the English-language literature, the first three monotherapy carnitine-treated hyperthyroid patients were reported in the mid-1960s [34]. The authors found that patients became clinically euthyroid without any consistent changes in the thyroid function tests, thus supporting the notion that the antithyroid effect of carnitine is one of peripheral antagonism of thyroid hormone, rather than a direct inhibition of thyroid gland function [35]. This was consistent with human tissue culture experiments where l-carnitine inhibited both cell entry and, to a greater extent, nuclear entry of both T3 and T4 [36]. These data are consistent with carnitine being a peripheral antagonist of thyroid hormone action, with a site of inhibition at or before the nuclear envelope [36].

The first controlled clinical trial addressing the value of l-carnitine in antagonizing elevated circulating levels of thyroid hormones was conducted in 50 women under Thyroid stimulating hormone (TSH)-suppressive l-T4 therapy for cytologically benign thyroid nodules who received a simultaneous treatment for six months with placebo (*n* = 10), or for given periods of time with l-carnitine (2g/d or 4 g/d to test dose-dependence) [37]. Evaluation by both extensive clinical and biochemical assessment demonstrated positive effects with the exception of osteocalcin, which increased further during l-carnitine administration and partial exception of total cholesterol (minimal or no increase during l-carnitine administration). Serum FT3, FT4 and TSH remained unchanged throughout the 180 day-duration of the trial. Thus, there was no antagonism from l-carnitine on the negative feedback that thyroid hormones exert on thyrotropin releasing hormone (TRH)/TSH. In addition to the hypothalamic TRH-producing neurons and the pituitary thyrotropin, also osteoblasts were refractory to the thyroid-hormone antagonizing effect of l-carnitine (see above). Thus, l-carnitine synergized with thyroid hormone on the osteoblasts to increase osteocalcin serum concentrations. The favorable effect on the osteoblasts was supported by measuring femur and lumbar bone density by dual-energy-X-ray absorptiometry [37].

More recent cases of severe forms of Graves’ disease-related hyperthyroidism, including thyroid storms, were treated successfully with l-carnitine [38,39,40]. Recently, a pilot study indicated the beneficial effects of a combination of l-carnitine and selenium supplementation in subclinical hyperthyroidism [41]. A rationale for a beneficial effect of l-carnitine supplementation in hyperthyroid patients seems likely because increased levels of thyroid hormones deprive the tissue deposits of l-carnitine itself [42], which is further substantiated by the finding of decreased concentrations of carnitine in the skeletal muscles of hyperthyroid patients. Interestingly, trendwise decreased concentrations of carnitine were found in skeletal muscles of hypothyroid patients [43], which were restored upon regaining euthyroidism. Therefore, decreased concentrations of carnitine in skeletal muscles may contribute to myopathy associated with either hypothyroidism or hyperthyroidism.

Sixty thyroid-hormone adequately replaced hypothyroid Korean patients (age 50.0 ± 9.2 years, 57 females) continued to complain of fatigue [44]. These patients were given l-carnitine (990 mg l-carnitine twice daily; *n* = 30) or placebo (*n* = 30) for 12 weeks. After 12 weeks, although neither the fatigue severity score nor the physical fatigue score changed significantly after 12 weeks, but the mental fatigue score was significantly improved by treatment with l-carnitine compared with placebo (*p* < 0.01). In subgroups, both the physical and mental fatigue scores improved significantly in patients younger than 50 years and those with free T3 ≥ 4.0 pg/mL by treatment with l-carnitine compared with placebo. Other case-based studies have indicated a benefit from l-carnitine on hypothyroid symptoms, but all of them have been case-based [45], while other studies may support benefits in the corticosteroid hormone setting [46].

## 3. Inositol: Compound and Physiology

Inositol is a water-soluble compound closely associated with the vitamin B group (also known as vitamin B8) [47]. Inositol is a carbohydrate which has a taste half as sweet as that of sucrose. Inositol has long been known for its metabolic effects in humans, where it plays a part in the synthesis of secondary messengers within cells. It is an essential component of the phospholipids that makes up cellular membranes and is found in virtually all cells. The most abundant form of the nutrient is myo-inositol. It assists in the transmission of nerve signals, helps to transport lipids within the body, and is also critical for the proper action of insulin and maintenance of cellular calcium balance. Foods containing the highest concentrations of myo-inositol include fruits, beans, grains and nuts. However, in grains, it is in a non-available form called phytate. The more bioavailable form of inositol comes from lecithin. Inositol is a necessary component of all cellular membranes. It is a member of the B-vitamin family that contributes to muscular and nerve function and participates in the metabolism of fats in the liver. Myo-inositol is the most abundant form of this nutrient, with its highest concentrations being found in the brain and central nervous system. Myo-inositol in particular is a versatile nutrient for the promotion of emotional and mental wellness, healthy eating patterns, and restful sleep through its critical role in neurotransmitter messaging systems. In addition, it is an important nutritional element for the maintenance of ovarian health and normal blood sugar maintenance, especially in women. Inositol is a non-essential member of the B-complex family with dietary sources from both animal and plant foods. The form of inositol used in this product is myo-inositol, the most abundant form of this nutrient. Inositol is found in all cell membranes, with the highest concentrations in the brain and central nervous system, where it plays an important role in neurotransmitter signaling. Inositol is also critical for the proper action of insulin, lipid metabolism, and for the maintenance of cellular calcium balance. Inositol is a necessary component of all cellular membranes. It is a member of the inositols are marketed as beneficial nutraceutics for improving mood and for the treatment of polycystic ovary syndrome [20]. A significant difference in the myo-inositol content of available products, and there are no regulations to ensure homogenous quality and accuracy [20].

### Inositol and Thyroid Function/Autoimmunity

Inositols are essential for the signaling of hormones such as insulin, gonadotropins (follicle stimulating hormone [FSH] and luteinizing hormone [LH]), and TSH. In the thyroid, imbalances in the inositol metabolism can impair thyroidal hormone biosynthesis, storage and secretion [47]. TSH signaling is rather complex involving two different signal cascades. One branch of the signal cascade involves as second messenger cyclic AMP (cAMP), while another branch is inositol-dependent [48]. In a controlled trial, 48 women with autoimmune subclinical hypothyroidism were randomized to treatment with either selenomethionine alone or selenomethionine plus myo-inositol. The authors demonstrated that patients with autoimmune thyroiditis and subclinical hypothyroidism, treated with myo-inositol and selenomethionine, had a reduction of the increased TSH, which selenomethionine supplementation alone was not able to promote. However, the concentration of both thyroperoxidase and thyroglobulin autoantibodies (TPOAb and TgAb) declined in both groups [48]. In a subsequent study of 86 patients with Hashimoto’s thyroiditis and subclinical hypothyroidism, the same authors found that the administration of myo-inositol and selenomethionine for six months significantly decreased TSH, TPOAb, and TgAb concentrations, while at the same time enhancing thyroid hormones and personal wellbeing, thereby restoring euthyroidism in patients diagnosed with autoimmune thyroiditis [49]. This was confirmed in a larger study of 168 patients with Hashimoto’s thyroiditis and subclinical hypothyroidism (TSH 3–6 mU/L) [50].

The mechanism of this effect might be through immune modulation rather than through thyroid function *per se* [51]. Using the afore-mentioned combined treatment in 22 patients with autoimmune thyroiditis, the initial TSH levels in the high normal range (2.1 < TSH < 4.0) significantly declined, suggesting that the combined treatment can reduce the risk of progression to hypothyroidism in subjects with autoimmune thyroid diseases. Antithyroid autoantibody levels also declined and, moreover, the suspected immune-modulatory effect was confirmed by the finding that the concentration of the chemokine CXCL10 also declined. Studies are, however, awaited to extend the observations in a larger population, to evaluate the effect on the quality of life, and to study the mechanism of the effect on chemokines.

Very recently, thyroid nodular disease also seemed to improve after the combined treatment with myo-inositol and selenomethionine [52], but this also needs confirmation. Final data in this study was analyzed from 34 patients with subclinical hypothyroidism: in 76% of mixed thyroid nodules a significant reduction of their size was observed and 56% of them significantly regressed nodule stiffness following oral supplementation with the combined nutraceutics for six months. The mean number of mixed thyroid nodules shifted from 1.4 ± 0.2 to 1.1 ± 0.2 (*p* ≤ 0.05) and the TSH concentrations dropped from 4.2 ± 0.2 mIU/L at baseline to 2.1 ± 0.2 mIU/L post-treatment (*p* < 0.001). In the control group, 38% of the thyroid nodules reduced their diameter but TSH concentrations significantly increased up to the threshold after six months (from 4.0 ± 0.2 mIU/L to 4.3 ± 0.2 mIU/L, *p* ≤ 0.05). However, further studies are required, both in vitro and in vivo, in order to investigate the mechanism of this effect on the one hand, and a possible clinical treatment use of myo-inositol plus selenomethionine for the general management of thyroid nodules on the other.

## 4. Melatonin: Compound and Physiology

The isolation of melatonin was first reported in 1958 [53]. Since the demonstration that pineal melatonin synthesis reflects both daily and seasonal time, melatonin has become a key element of chronobiology research. In mammals, pineal melatonin is essential for transducing day-length information into seasonal physiological responses. Due to its lipophilic nature, melatonin is able to cross the placenta and is believed to regulate multiple aspects of perinatal physiology. The endogenous daily melatonin rhythm is also likely to play a role in the maintenance of synchrony between circadian clocks throughout the adult body. Pharmacological doses of melatonin are effective in resetting circadian rhythms if taken at an appropriate time of day and can acutely regulate factors such as body temperature and alertness, especially when taken during the day. Despite the extensive literature on melatonin physiology, several key questions remain unanswered. Particularly the amplitude of melatonin rhythms has recently been associated with diseases such as type 2 diabetes mellitus but the physiological significance of melatonin rhythm amplitude remains poorly understood.

As a nutraceutical, melatonin is easily available over the counter and is marketed to regulate the sleep pattern and adaptation to time zone differences among numerous other conditions.

### Melatonin and Thyroid Function

Melatonin has antioxidant properties, which is one of the reasons why it is assumed to be beneficial for many disease conditions. However, very few human studies exist, and they are primarily of a physiological nature. One such study considers several endocrine and immune interactions in healthy persons at different ages [54] and found statistically significant time-qualified correlations among lymphocyte subset percentages and hormone serum levels in the young and middle aged and one could speculate that the phenomenon of lymphocyte subpopulation redistribution may be more complex, and may involve other hormones such as TRH, TSH, GH (growth hormone), IGF1 (insulin-like growth factor 1), monoamines such as melatonin, cytokines such as IL2 (Interleukin 2), and chemokines. The aging of immune system function may be related to the alteration of circadian rhythmicity, with a loss of interaction among key lymphocyte subsets, immunomodulating hormones, as well as cytokines/chemokines.

Thirty-six perimenopausal and 18 postmenopausal women between 42 and 62 years of age with no pathology or medication were selected for a randomized study of melatonin or placebo at bedtime (22:00–00:00). The melatonin concentration was measured in saliva to divide the participants into low, medium, and high-melatonin subjects [55]. Three- and six-months later, blood was taken for the determination of pituitary (LH and FSH), ovarian, and thyroid hormones (T3 and T4). The results showed that women low in melatonin after treatment with melatonin significantly increased thyroid hormones levels and improved gonadal functions [55]. These results were confirmed by the same authors in another study where peri- and menopausal women (*N* = 139) took a daily dose of 3 mg synthetic melatonin or placebo for 6 months. Melatonin concentrations were determined from five daily saliva samples at fixed times while other hormone levels were determined from blood samples three times over the six-month period [56]. The conclusion was that the six-months treatment with melatonin produced a remarkable and highly significant improvement of thyroid function, positive changes of gonadotropins towards more juvenile levels, and the abrogation of menopause-related depression.

In 40 menopausal women the combination of myo-inositol plus melatonin seemed to positively affect glucose metabolism. Myo-inositol alone seemed to improve thyroid function, while addition of melatonin increased the serum TSH concentration [57]. The reason for this is unknown, but all melatonin products warn against worsening of autoimmune diseases on basis of its potential effect on the immune system. Recently, SNPs related to melatonin receptor gene polymorphism haplotypes were associated with susceptibility to Graves’ disease in an ethnic Chinese population and thus support the involvement of the melatonin pathway in the pathogenesis of this autoimmune thyroid disease [58].

In conclusion, there is to date no controlled trials to substantiate a use of melatonin for general thyroid health improvement. 

## 5. Resveratrol: Compound and Physiology

Resveratrol (3,4′,5-trihydroxy-trans-stilbene) belongs to the flavonoids family and is a major natural polyphenolic compound found in several fruit and vegetables such as grapes, peanuts, and peanut sprouts. It seems to play an important role as a therapeutic and chemopreventive agent used in the treatment of various illnesses [59,60] and has therefore recently gained much attention among health professionals as well as other nutrition experts. Resveratrol exhibits effects against several cancers [61,62] through different pathways and, furthermore, it has antidiabetic, anti-inflammatory, and antioxidant effects. The cardiovascular protective capacities of resveratrol are believed to be associated with multiple molecular targets such as inflammation, oxidative stress, apoptosis, mitochondrial dysfunction, angiogenesis and platelet aggregation [59].

Similarly, resveratrol is a potent scavenger for free radicals. The high efficiency of resveratrol might be due to the three hydroxyl groups in its structure. Thus, the use of resveratrol as a health-promoting dietary supplement is rapidly increasing in today’s market. Many reports have shown that resveratrol offers a wide range of preventive and therapeutic alternatives against various diseases including different types of cancer.

Resveratrol is a member of a family of enzymes, under the general name of stilbene synthase, which makes up part of a larger family of proteins with numerous functions. Notably, its chemical structure resembles that of l-T4, however it is not clear if this has any functional implications [63]. Resveratrol synthase is developed from chalcone synthase via gene duplication and mutations. The absorption in humans is approximately 75% (delayed by food) by trans-epithelial diffusion, while tissue accumulation enhances efficacy at target sites.

### Resveratrol and Thyroid Function

Resveratrol may arrest the proliferation of thyroid cancer cells by increasing the abundance and phosphorylation of p53 [64,65,66]. Moreover, resveratrol mediates the regulation of TSH while, due to its effects on iodine trapping, it shows promise as a prospective anti-thyroid drug. On the other hand, these effects also resulted in a pronounced proliferative action on thyrocytes and resveratrol may therefore be a thyroid disrupting compound [67]. No clinical studies on the compound’s effect on the thyroid has been performed in humans, so all available evidence is based on animal and in vitro cellular studies.

Finally, resveratrol as an antioxidant agent is a free radical scavenger and this property can be of interest in thyroid disease states that are accompanied by increased production of hydrogen peroxide and radical oxygen species, such as autoimmune thyroiditis and hyperthyroidism [68]. Proper randomized clinical trials would, however, be required before implementing any use.

Resveratrol supplements can be easily purchased over the counter but they are not regulated by the FDA or any other health authority. Most resveratrol capsules sold in the U.S. contain extracts from an Asian plant called *Polygonum cuspidatum*. Other resveratrol supplements are made from red wine or red grape extracts. The dosages in most resveratrol supplements typically contain 250 to 500 milligrams, which is much lower than the amounts that have been shown beneficial in research (2000 milligrams of resveratrol or more a day).

## 6. Selenium: Compound and Physiology

Selenium is a non-metal chemical element that is an essential micronutrient. Selenium salts are toxic in large amounts, but trace amounts are necessary for cellular function in many organisms, including all animals. Dietary selenium comes from nuts, cereals, and mushrooms. Brazil nuts are the richest dietary source (though this is soil-dependent since the Brazil nut does not require high levels of the element for its own needs). Selenium is an ingredient in many multivitamins and other dietary supplements. It is a component of the antioxidant enzymes glutathione peroxidase and thioredoxin reductase, which indirectly reduce certain oxidized molecules in animals and some plants. It is also found in three deiodinase enzymes, which convert one thyroid hormone to another. In living systems, selenium is found in the amino acids selenomethionine, selenocysteine, and methylselenocysteine.

The U.S. recommended dietary allowance (RDA) for teenagers and adults is 55 µg/day. Selenium as a dietary supplement is available in many forms, including multi-vitamins/mineral supplements, which typically contain 55 or 70 µg/serving. Selenium-specific supplements typically contain either 100 or 200 µg/serving. In June 2015, the U.S. FDA published its final rule establishing the requirement of minimum and maximum levels of selenium in infant formula. The reference values of EFSA for selenium range from 15 µg/day for children aged one to three years to 70 µg/day for adolescents aged 15–17 years [69]. The selenium content in the human body is believed to be in the range of 13–20 milligram [70].

Selenium food supplements are most efficient as yeast-based selenomethionine, but the contents are not standardized or under any control. For instance, six different brands of yeast-based selenium food supplements were analysed for the expected selenomethionine content [23]. Only two brands had high levels of selenomethionine; one brand appeared to contain only inorganic selenium, and one brand appeared to contain more than half inorganic selenium despite label claims of content being only selenomethionine. Nevertheless, selenium supplementation is increasingly prescribed by endocrinologists as recently documented for Italian endocrinologists [70]. In detail, approximately one in four respondents use selenium often/always, with only one in either use never. Rates were approximately one-fourth of respondents prescribing selenium often/always in Hashimoto’s thyroiditis, and one-fifth prescribing selenium in the case presented. In patients with autoimmune thyroiditis (AIT) who are planning pregnancy or are already pregnant, approximately 40% of respondents suggest selenium use [71]. It is worth underlining that the American Thyroid Association (ATA) pregnancy guideline reported that “selenium supplementation is not recommended for the treatment of TPOAb-positive women during pregnancy” [72].

### Selenium and Thyroid Function/Autoimmunity

Among all tissues, the thyroid gland has the highest concentration of selenium, of which much is stored in the thyrocytes as the selenoproteins [73,74]: deiodinases (DI1, DI2), glutathion peroxidase (GPx1, GPx3, GPx4), and thioredoxin reductases (TR1, TR2). Both the thyroid gland and all other cells that are dependent of thyroid hormone for proper function use selenium as a cofactor for three of the four known types of thyroid hormone deiodinases, which can both activate and deactivate thyroid hormones and their metabolites—the iodothyronine deiodinases are the subfamily of deiodinase enzymes that use selenium, as does the otherwise rare amino acid selenocysteine. Only iodotyrosine deiodinase does not use selenium.

Adequate selenium intake is required for normal function of thyrocytes and the angiofollicular units in thyroid hormone biosynthesis and storage. Inadequate selenium intake has been associated with increased thyroid volume in females, but not males in one study [75], and in a larger Danish population, this negative correlation between selenium status and thyroid volume was confirmed, and there was, furthermore, a trend toward increased numbers of thyroid nodules with inadequate selenium status [74,76]. Adequate selenium intake, with respect to proper thyroid function, can be monitored by the analysis of serum or plasma selenoproteins such as selenoprotein P or plasma GPx3 [74,77,78]. Intoxication has been reported in several places in China from dietary intake and soil contamination [79,80]. Measurement of these variables is becoming more important in the view of the increased interest in selenium supplementation in various patient groups particularly with autoimmune thyroid diseases (see below) and since there is a risk of overdosing by general too high doses on the one hand and supplementation of selenium sufficient individuals on the other. The U-shaped curve of beneficial effects from selenium concentrations, i.e., exhibiting major advantages in selenium-deficient individuals but specific health risks in those with selenium excess should be seriously considered [81].

Selenium status has been shown to affect immune functions, e.g., T cell differentiation, and selenium deficiency has been associated with Th2 cells/markers, while higher selenium concentrations seem to favor an increased Th1 and Treg response [82]. These observations are thus in keeping with the suggestion of beneficial effects of selenium supplementation in autoimmune diseases of the thyroid [73,83]. Newly diagnosed autoimmune hyperthyroidism, Graves’ disease, has been associated with low selenium concentrations [84], an observation which has fuelled several interventional treatment studies of selenium supplementation as adjunctive to antithyroid drugs in Graves’ disease [85,86,87,88]. A very recent systematic review and meta-analysis of 10 randomized clinical trials could not substantiate a systematic effect of selenium supplementation as an adjunctive treatment in Graves’ disease [89]. Generally, the studies were all underpowered, of too short a duration, and with too broad clinical characteristics of the patients, and the issue is therefore yet to be resolved—results from larger ongoing prospective studies are awaited [90].

Concerning the subpopulations of Graves’ disease, however, a prospective case-control study demonstrated lower serum selenium concentrations in patients with Graves’ orbitopathy compared to Graves’ patients without orbitopathy in an Australian study population with marginal selenium status [91]. Against this background, relative selenium deficiency may be an independent risk factor for orbitopathy in patients with Graves’ diseases. This has been further substantiated by one major multicentric prospective, placebo and serum-controlled study of Graves’ patients with orbitopathy, with demonstration of improved quality of life and disease activity scores [92].

Several placebo-controlled and double-blind studies, both observational and prospective, have been performed to demonstrate the improved quality of life, wellbeing, thyroid hormone status, and disease symptoms of chronic autoimmune thyroiditis of the Hashimoto type with or without hypothyroidism. Although many studies have consistently demonstrated a reduction in thyroid autoantibody concentrations by selenium supplementation, including some compared with control/placebo [93,94,95,96], recent meta-analyses found insufficient evidence for the clinical efficacy of selenium supplementation in chronic autoimmune thyroiditis [97,98]. Hopefully, future trials can ultimately provide reliable evidence to help inform clinical decision making. Results were less optimistic than the individual study results, many of which were, however, underpowered, and therefore, in this autoimmune patient group, results are unclear and further ongoing study results are awaited [99].

In women at risk of postpartum thyroiditis, adequate selenium status prevents its development. In a prospective placebo-controlled double-blind prevention study [100], there were fewer cases of postpartum thyroiditis—these results, however, have not been confirmed in other studies [73,101].

Finally, there has been no indication of an increased risk of thyroid cancer in either selenium deficiency or with supplementation of selenium [74].

In conclusion, selenium status has a high impact on normal thyroid development and function, and it is still a potential candidate for improvement of clinical markers and quality of life in some situations of autoimmune thyroid diseases by supplementation, e.g., Graves’ orbitopathy and possibly postpartum thyroiditis. However, more solid evidence is awaited until firm conclusions can be made concerning recommendations for global routine clinical use.

## 7. Perspective and Conclusions

As clinicians, we often see patients who are taking all sorts of supplements with the hope of improving their health and medical conditions, as well as simply feeling better.

Thyroid supplements attract a disproportionately large amount of attention, just as the thyroid gland gets “blamed” for multiple symptoms. There are truths and myths that this review had tried to clarify. Of the numerous nutraceuticals out there for thyroid disease management, we focused on the common or popular ones we encounter in the clinical practice.

Clinicians should acknowledge that over 30% of our patients are using supplements and thus should inquire about them during our office encounters. Apart from improving their general health, patients are using these alleged thyroid supplements to help “improve their metabolism, have more energy, and to lose weight”.

It is important that we do not just dismiss these patients, but rather have honest discussions about the claimed benefits and potential risks. Physicians would do well to familiarize themselves with the main supplements being used, and also to know the scientific evidence available to support or refute these claims. More importantly, physicians should understand the potential risks or side effects in order to properly counsel patients about their use.

Based on the literature reviewed in the preceding sections, the evidence for the clinical use and potential benefit of the nutraceuticals addressed in this paper is summarized in Table 4. It is, however, worth noting that very few studies have been randomized clinical trials and generally all the studies have lacked proper power and even attempts to perform power calculations including the few randomized clinical trials. For selenium, two randomized, properly powered, placebo controlled clinical trials are ongoing and results are awaited [89,98]. Similar studies are required also for the most relevant nutraceuticals with a possible influence on the thyroid, in order to provide proper guidance both to patients and clinicians.

## Figures and Tables

**Table 1 nutrients-11-02214-t001:** Some definitions of “nutraceutical”.

Reference	Definition
[1]	“A foodstuff (such as a fortified food or dietary supplement) that provides health benefits in addition to its basic nutritional value. (First known use: 1990)”.
[2]	“A food to which vitamins, minerals, or drugs have been added to make it healthier.”
[3]	“Nutraceuticals, which have also been called medical foods, designer foods, phytochemicals, functional foods and nutritional supplements, include such everyday products as “bio” yoghurts and fortified breakfast cereals, as well as vitamins, herbal remedies and even genetically modified foods and supplements. Many different terms and definitions are used in different countries, which can result in confusion.”
[4]	“I propose to redefine functional foods and nutraceuticals. When food is being cooked or prepared using “scientific intelligence” with or without knowledge of how or why it is being used, the food is called ‘functional food’. Thus, functional food provides the body with the required amount of vitamins, fats, proteins, carbohydrates, etc., needed for its healthy survival. When functional food aids in the prevention and/or treatment of disease(s) and/or disorder(s) other than anemia, it is called a nutraceutical.”
[5]	Nutraceutical combines two words the term ‘nutrition/nutrients’ (a nourishing food component) and ‘pharmaceutical’ (medicine or a substance used as a medication) applied to food or food component products sometimes with active principle from plants that can provide health and medical benefits, including the prevention and treatment of disease.

**Table 2 nutrients-11-02214-t002:** Summary of number of articles on given nutraceuticals retrievable on PubMed as of 9 July 2019.

	Entry	No. of Items		Proportions		
		Total	Human	Human/Total	Thyroid/Total	Thyroid/Human
1	nutraceuticals	81,422	52,406	64.4%	N/A	N/A
2	nutraceuticals AND hormones	5698	3664	61.4%	N/A	N/A
3	nutraceuticals AND thyroid	656	487	74.2%	0.8%	0.9%
4	carnitine	16,737	7831	46.8%	N/A	N/A
5	carnitine AND thyroid	145	68	46.9%	0.9%	0.9%
6	inositol	44,801	16,700	37.3%	N/A	N/A
7	inositol AND thyroid	295	141	47.8%	0.6%	0.8%
8	melatonin	24,921	10,740	43.1%	N/A	N/A
9	melatonin AND thyroid	514	195	37.9%	2.1%	1.8%
10	resveratrol	11,983	5447	45.4%	N/A	N/A
11	resveratrol AND thyroid	78	47	60.2%	0.6%	0.9%
12	selenium	33,980	13,333	39.2%	N/A	N/A
13	selenium AND thyroid	938	576	61.4%	2.8%	4.3%

Note that the number of items under the keyword “nutraceuticals” underestimates the bulk of the literature. Indeed, by adding items #4, 6, 10, 12, 14 and 16 the sum is 164,513, which is greater than 81,422 for item #1. Similar considerations apply for the corresponding human studies (67,565 vs. 52,406), and for the thyroid studies (total studies = 2305 vs. 656; human studies = 1269 vs. 487).

**Table 3 nutrients-11-02214-t003:** Economic issues for the reviewed nutraceuticals.

Nutraceutical	Market and Sales ^
l-carnitine	l-carnitine market is expected to be worth USD 127 million by 2017, with the United States being the largest market, and the Asia-Pacific region, particularly China, expected to experience a 5.5% annual growth rate through 2017 [12]. No. of items on sale-Amazon: 53; Walgreens: No match; CVS Pharmacy:13.
Myo-inositol	In the consumption market, the global consumption value of inositol increases with the 2.01% average growth rate. Europe and China are the mainly consumption regions [11]. With myo-inositol being the most common form of inositols, over the next five years the inositol market, will register a 6.8% compound annual growth rate in terms of revenue, the global market size will reach US $140 million by 2024, from US $94 million in 2019 [13]. No. of items on sale-Amazon: 3; Walgreens: No match; CVS Pharmacy: No match.
Melatonin	The North America region is the largest supplier of melatonin, with a production market share nearly 54% in 2016, Europe coming next with 27% [14]. The global market size will reach US $2080 million by 2024, from US $700 million in 2019 [14]. No. of items on sale-Amazon: 122; Walgreens: 11; CVS Pharmacy: 91.
Resveratrol	Resveratrol supplements, with annual sales of $30 million in the United States [15] No. of items on sale-Amazon: 45; Walgreens: No match; CVS Pharmacy: 19.
Selenium	Selenium market reached $87 million U.S. in 2017 [16]. No. of items on sale-Amazon: 91; Walgreens: No match; CVS Pharmacy: 84.

^ Numbers in brackets are references. Internet sales by Amazon, Walgreens and CVS Pharmacy are reported. Search was performed for the pure nutraceutical, such as entering “pure melatonin”. Search performed on the Amazon website by omitting the word “pure”, yielded a greater number of results (973 for l-carnitine, 48 for myo-inositol, 178 for resveratrol, over 1000 for selenium, and over 1000 for biotin). Search on the Walgreens website by omitting the word “pure”, yielded a greater number of results (12 for l-carnitine, 1 for myo-inositol, 107 for melatonin, 10 for resveratrol, 21 for selenium, and 71 for biotin).

**Table 4 nutrients-11-02214-t004:** Summary of evidence for clinical use of the nutraceuticals reviewed here in the thyroid setting *.

Question: Is There Evidence for Clinical Use of …?	Answer
Carnitine	Currently available evidence supports the usefulness of l-carnitine in hyperthyroid patients. Carnitine ameliorates a number of symptoms and signs, including cardiac arrhythmia. Case reports have shown benefits even in the setting of thyroid storm. However, no changes in thyroid function tests were reported. One practical setting for the use of l-carnitine (two grams per day) is the control of hyperthyroidism symptomatology when the patients need to take low doses of antithyroid drugs. Only one Korean study is currently available for hypothyroidism, thus precluding conclusions.
Inositols	Only in one study, MI alone (2 g twice a day) or MI plus melatonin (2 g/d MI plus 3 g/d melatonin) were given in two groups of euthyroid postmenopausal women, and serum FT4 and TSH evaluated. MI alone caused an almost 3.5% increase in serum FT4 and a 10% decrease in serum TSH. This contrasted with the opposite changes (3.5% decrease in serum FT4 and almost 10% increase in serum TSH) observed in the group under MI plus melatonin. Few studies have been conducted only in one Western country (Italy), and with the combination of MI plus selenium or MI plus carnitine. Supplementation with the first combination has been used in the setting of patients with Hashimoto’s thyroiditis related SCHypo, and it decreased both serum thyroid autoantibodies and TSH. The combination of MI plus carnitine was only investigated in one study of patients with SCHyper, thus precluding conclusions.
Melatonin	There has been interest in melatonin and autoimmunity and the thyroid gland has been implicated in the discussion. It is thought that melatonin may have a paracrine role and in thyroid disease under a condition of oxidative stress may reduce the processes involved in thyroid antoimmunity. However, there are no controlled trials or definite data to show conclusively that melatonin can be beneficial in thyroid disease.
Resveratrol	No answer can be given, simply for lack of studies.
Selenium	Benefits have been demonstrated for mild forms of Graves’ ophthalmopathy. Benefits for the clinical course of GD itself are controversial. In the setting of HT, a benefit has been shown more on serum thyroid autoantibodies than on thyroid function. There is only one study on the benefit given by selenium supplementation, both in terms of serum thyroid autoantibodies and thyroid dysfunction, in the setting of PPT. For the combinations of selenium with MI see above.

Abbreviations, in alphabetical order: GD = Graves’ disease; HT = Hashimotos’ thyroiditis; MI = myo-inositol; PPT = postpartum thyroiditis. SCHyper = subclinical hyperthyroidism; SCHypo = subclinical hypothyroidism.

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
