# Peer review of "Nutraceutical Supplements in the Thyroid Setting: Health Benefits beyond Basic Nutrition"

_nutrients, 2019, doi:10.3390/nu11092214_

Round 1

Reviewer 1 Report

The manuscript is fluent and well written. The deepening of the use of over-the-counter supplements, especially in the field of thyroid disorders, is a topic of great interest that could stimulate future research in this field. The section on purity and concentration of over-the-counter products is also very interesting and actual.

However, there remain some aspects that should be improved and that currently make the manuscript weak and ambiguous in some passages.

The most critical aspect is the nutraceutical concept which, despite the attempt to provide various definitions, remains ambiguous and difficult to apply. While taking into consideration the first definition of Dr De Felice, perhaps the only one that should be highlighted, it is not clear whether these are products with a nutritional effect or with pharmacological interest, so they can be counted neither among the supplements nor among the drugs. We cannot think that food supplements are exempt from pharmacological effects but the ambiguity of the concept of nutraceutical, at the edge between food and medicine, creates strong confusion. I realize that the whole manuscript is focused on the idea of ​​nutraceutical and therefore it is unthinkable to purge it from this concept, however, the question should be treated with more precaution.

In my opinion, the substances indicated are not necessarily definable as nutraceuticals while some plant extracts, and their naturally contained active principles, are much closer to the concept of nutraceutical. Since there is no one single shared definition, it remains a speculation. The feeling is that the terms supplement and nutraceutical are used along with the manuscript interchangeably.

- For example, given the European origin of 3 out of 4 Authors, EFSA's position on the concept of nutraceutical cannot be ignored. A reference should be made to the European directive which establishes the field of application of food supplements that cannot claim prevention or cure and therefore that exclude a priori the concept of nutraceutical, which in fact, is not recognized in Europe.

- Similarly, the Japanese legislation concerning FOSHU, the official concept closest to the nutraceutical, cannot be omitted.

- The authors could adopt a clearer definition, considering that the proposed definitions are already contradictory, highlighting the necessary precautions, to clarify the method with which the substances discussed were selected.

- More generally, it seems that the choice of substances to be treated in the manuscript derives from a personal selection. For example, why not mention vitamin D and its immunomodulatory power which could have a role in autoimmune thyroid disease?

- In the abstract, too much interest has been devoted to biotin, almost suggesting that the manuscript could present a more specific focus on this vitamin. Lines 25 to 28 could be omitted or at least resized.

- Table 1 helps to generate confusion. It should be resized. Alternatively, it could be converted in a discursive way as a real paragraph. The definition adopted by the authors must be clear.

- The lines 72 to 81 are offtopic, as the authors themselves point out. They do not help to clarify the concept of nutraceutical but only the possible metabolic interaction of a substance.

- Regarding the physiology of carnitine, cardiac function should also be mention.

- At line 325, it is always a good idea to specify in extenso the first time an abbreviation is used.

- At line 382, ​​also the dietary reference value of EFSA must be specified.

- The reference for the selenium content in the human body on lines 386-387 must be added-

- The arguments at lines 388-393 have already been discussed at lines 140-145.

- Lines 465-469 have already been discussed at lines 394-399.

- At line 474, please, specify the reference from which dietary recommendations came from. If it is a US reference, specify European ones as well.

- At line 477, please, specify the bibliographic reference.

- More generally, a risk/benefit balance should be investigated for each substance, highlighting potential toxicity risks or collateral physiological effects, especially when non-physiological concentrations are used.

- Table 1 of Appendix A can be enhanced with EFSA references in addition to the FDA one as well. In any case, it contributes to generating confusion, because it is referred to supplements and not to nutraceuticals. Lines 39-41 imprecisely indicate that the table also refers to nutraceuticals.

Reviewer 2 Report

This is a review describing the effects of dietary supplements including L-carnitine, myo-inositol, selenium, melatonin, and resveratrol on thyroid function, and the effects of biotin on thyroid function tests. This is not a topic which has been previously reviewed, and this paper does have clinical relevance.

Overall the manuscript seems overly long and diffuse. I would suggest removing the section on biotin (since this does not actually affect thyroid function, and the effects on thyroid function tests have recently been reviewed elsewhere). I would also try to shorten overall by about 20% and to make sure that for each section there are concluding statements, not simply long descriptions of (often contradictory) studies without summation. Throughout the manuscript it should be clarified regarding any discussion of supplement regulations or market share which region/country is being discussed. Tables: the center justification of all of the tables makes them difficult to read. Lines 52-59: the discussion of the change in the numbers of papers between 2017 and 2019 could be removed. Lines 74-81: the discussion of ginseng is not really pertinent and could be removed. Line 101: is there a more relevant reference here than #8? The issue of purity: in this section it is worth noting the illegal presence in many US supplements marketed specifically for thyroid health: Kang GY et al. Thyroxine and triiodothyronine content in commercially available thyroid health supplements. Thyroid. 2013;23(10):1233-7. Lines 312-14, “The results showed that pituitary and thyroid functions recovered… towards a more juvenile pattern…”: this is unclear as written. Lines 329-31: melatonin “may be tried for specific purposes”: what purposes? This needs to be clearer. Lines 398-99: with regard to use of selenium supplements for pregnant women with thyroid autoimmunity,. It should be noted that the 2017 ATA pregnancy guideline specifically recommends against this. Lines 463-469: this repeats data cited earlier in the section and could be removed.

Round 2

Reviewer 1 Report

Requests have been sufficiently met; however, only little aspects remain to be reviewed.

- In Table 1, it would be desirable to delete the definition that emerges from references 5 and 6 because it does not imply a definition of nutraceutical but simply of dietary supplements, already discussed and clarified in the appendix.

- Line 92 can be removed, taking into account that the previous part has been omitted.

- I could not identify the reference value of EFSA for selenium in the manuscript (look at Https://efsa.onlinelibrary.wiley.com/doi/epdf/10.2903/j.efsa.2014.3846).

- Please, check the reference used on line 387. Perhaps, the authors refer to the EFSA opinion "10.2903 / j.efsa.2016.4368". I advise using it with caution because it could refer to to the synthetic trans-resveratrol. To avoid confusion, the removal of the sentence from lines 386-387 would be desirable.

Reviewer 2 Report

This revised version is more focused and easier to read, but a few additional clarifications are still needed.

1. Market and sales, lines 94-99: does the $23 billion figure refer to US or global sales? What about the $278 billion dollar estimate?

2 . Lines 228-232: this is a run-on sentence and hard to understand as written.

3. Lines 324-327: what was the sample size in the study described

4. Lines 465-470: another run-on sentence; difficult to understand.

5. Table 4: Carnitine, "the current evidence supports old studies in hyperthyroid patients": unclear as written.

6. Table 4, Inositols: could this table entry be shortened and simplified? It is difficult to follow as written.
